

# State of the art and future directions for measuring event-related potentials during cycling exercise: a systematic review

Rémi Renoud-Grappin[1], Lionel Pazart[1,2], Julie Giustiniani[1,2,3] and Damien Gabriel[1,2]

[1] UMR INSERM 1322 LINC, Université de Franche-Comté, Besançon, France
[2] Inserm CIC 1431, Centre Hospitalier Universitaire, Besançon, France
[3] Service d'addictologie, Centre Hospitalier Universitaire, Besançon, France

## ABSTRACT

**Intro:** Electroencephalography (EEG) is a technique for measuring brain activity that is widely used in neuroscience research. Event-related potentials (ERPs) in the EEG make it possible to study sensory and cognitive processes in the brain. Previous reports have shown that aerobic exercise can have an impact on components of ERPs such as amplitude and latency. However, they focused on the measurement of ERPs after exercise.

**Objectives:** The aim of this systematic review was to investigate the feasibility of measuring ERPs during cycling, and to assess the impact of cycling on ERPs during cycling.

**Methods:** We followed the PRISMA guidelines for new systematic reviews. To be eligible, studies had to include healthy adults and measure ERPs during cycling. All articles were found using Google Scholar and by searching references. Data extracted from the studies included: objectives of ERP studies, ERP paradigm, EEG system, study population data, exercise characteristics (duration, intensity, pedaling cadence), and ERP and behavioral outcomes. The Cochrane Risk of Bias 2 tool was used to assess study bias.

**Results:** Twenty studies were selected. The effect of cycling on ERPs was mainly based on a comparison of P3 wave amplitude between cycling and resting states, using an attentional task. The ERP paradigm most often used was the auditory oddball task. Exercise characteristics and study methods varied considerably.

**Discussion:** It is possible to measure ERPs during cycling under conditions that are likely to introduce more artifacts, including a 3-h athletic exercise session and cycling outdoors. Secondly, no assessment of the effect of cycling on ERPs was possible, because the methods differed too widely between studies. In addition, the theories proposed to explain the results sometimes seemed to contradict each other. Although most studies reported significant results, the direction of the effects was inconsistent. Finally, we suggest some areas for improvement for future studies on the subject.

Corresponding authors
Rémi Renoud-Grappin,
remirenoudgrappin@gmail.com
Lionel Pazart,
lionel.pazart@univ-fcomte.fr

## INTRODUCTION

Electroencephalography (EEG) was the first non-invasive method to directly measure brain activity, in the early 1930s (*Berger, 1931*). EEG is still widely used today, and captures the micro-currents resulting from electrical activity generated by ionic flows as neurons connect and fire (*Bear, Connors & Paradiso, 2016*). The change in electrical potential produced by the nervous system in response to internal or external stimuli is known as event-related potential (ERP). Internal stimuli are linked to cognitive activity (attention, memory, motor preparation, *etc.*), while external triggering events are sensory stimuli (visual, auditory, *etc.*). Evoked potentials can therefore be used in both clinical practice, to verify the proper functioning of the brain, and in neurophysiological research, to understand the functional organization of the nervous system.

As these are micro-currents (in the microvolt range), it is often necessary to repeat the recording a large number of times to reliably characterize an ERP by increasing the signal-to-noise ratio. EEG signal "averaging" made it possible to use ERPs in neuroscience research from the 1960s onwards (*Woodman, 2010*). The evoked potential can be described in terms of different parameters including its amplitude, latency, *etc.*

One advantage of ERP measurement over other techniques for measuring brain activity is its precision or temporal resolution, since it makes it possible to observe brain responses that take place over a few tenths of a second, with no conduction delay in relation to actual brain activity. However, with this method, it is difficult to distinguish the precise location of the neural generators of the ERPs measured (*Woodman, 2010*), although new approaches based on high-resolution EEG are improving accuracy (*Kristeva-Feige et al., 1997*). Spatial resolution remains inferior to other techniques for measuring brain activity, such as fMRI.

In response to an internal or external stimulus, the simultaneous activity of a large number of neurons oriented perpendicular to the scalp surface generates a wave called an "ERP wave", which can be detected by EEG. An ERP wave has several components. These include its polarity (N for negative, P for positive), amplitude and latency. We can also describe its distribution on the scalp (fronto-parietal, for example) and its sensitivity to a type of stimulus. Some ERP components are linked to specific components of executive function, such as the speed of the attentional process, reflected by ERP latency, or the quantity of cognitive resources, reflected by ERP amplitude.

ERP measures have been linked to low-level sensory processes, such as perception, and to cognitive processes, such as attention, inhibition, response choice, error feedback processing, memory activity and other cognitive functions (*Helfrich & Knight, 2019*). The most widely studied ERP is the P3 wave, which represents a family of ERPs with positive deflection and for which the wave appears around 250 to 500 milliseconds after the stimulus (around 300 ms). The discovery of the P300 goes back a long way. As early as 1964, during a study of evoked potentials, *Chapman & Bragdon (1964)* noticed a variation in brain activity around 300 ms after the presentation of stimuli with which the subjects were unfamiliar. The P3 is considered to be an endogenous potential, as its occurrence is not linked to the physical attributes of a stimulus, but rather to the individual's reaction to

it. Depending on the experimental protocol, this wave can be interpreted as an index of attention to a target, including its detection or discrimination. In addition, it can be used as a marker of attentional performance (accuracy, response time) (*Helfrich & Knight, 2019*). Its amplitude component can even be influenced by the valence (gain/loss) of a feedback stimulus, demonstrating its link to high-level cognitive processes. Prolonged P300 latency and reduced amplitude indicate difficulties in processing and responding to infrequent target stimuli, which are crucial for attention and cognitive control (*Polish & Kok, 1995*). The procedure by which the P300 wave is most commonly studied is the oddball paradigm. The first article citing the discovery of the P300 using the oddball paradigm is *Sutton et al. (1965)*. In this test, the subject is presented with at least two different stimuli: one is the non-target item (which appears frequently), the other is the target item (its appearance is rarer and requires a reaction from the subject).

One recent application of ERPs is to explore variations in neural activity during exercise. Aerobic exercise calls on brain resources to execute a movement or maintain a physical effort. The theories of *Dietrich (2006)* and *Dietrich & Audiffren (2011)* assume a transfer of frontal neural resources to the aerobic effort regions that need them. Several studies have shown that after aerobic exercise, the amplitude and latency of ERPs are affected (*Gusatovic et al., 2022*). Moderate exercise intensity affects ERPs the most, suggesting the importance of exercise intensity on ERPs. It remains unknown whether the effects of exercise or exercise intensity on ERPs differ when measured during or after aerobic exercise. Thus, the study of ERPs during exercise may open up new perspectives in the study of the direct impact of aerobic exercise on cognitive functioning. Measuring brain activity during movement presents obstacles (*Thompson et al., 2008*), but great progress has been achieved in terms of signal processing (*Sadiya, Alhanai & Ghassemi, 2021*) and equipment. An increasing number of mobile EEG devices are appearing on the market, offering performances comparable to those of fixed devices (*Chabin et al., 2020*). Moreover, specific conditions of exercise are adapted to the measure of ERPs. Cycling on a bicycle or cycloergometer is the type of exercise most often used for EEG measurements during aerobic exercise, because the head remains relatively immobile. Compared to treadmill running for example, the number of artifacts is considerably reduced.

The main aim of this review was to examine the feasibility of measuring ERPs during cycling, and to examine the methods used to perform these measurements. It also aimed to investigate the effect of pedaling on ERPs, in particular by comparing them according to exercise intensity.

## MATERIALS AND METHODS

We followed the PRISMA guidelines for new systematic reviews. The PRISMA guidelines relating to the synthesis of results as for meta-analysis were not applicable in this review because there were too few studies and methods were too heterogeneous, precluding meta-analysis. We used the Cochrane risk-of-bias tool 2 (*Sterne et al., 2019*) to assess the risk of bias of the studies included.

## Selection criteria

We chose to include studies involving adults only, as the effect of exercise on ERP in children may be different and would introduce bias into the results. For the same reason, we excluded studies that included pathological conditions. In addition, we chose to include only studies in which the mode of exercise was pedaling (on a cycloergometer or bicycle), in order to reduce variability in methods between studies, thus rendering the results more comparable.

## Search equation

A first search was performed on Google Scholar in June 2023 with the following equation: "event-related potentials" AND (cycling OR pedaling OR biking). To avoid excluding potentially relevant articles, no filters or time limits were used. The search for studies had to be carried out using a broad, unfiltered search equation on Google Scholar, which offers articles that are less relevant overall, but more numerous. With PubMed, this equation yielded only one result, not relevant to this review. Web of Science did not identify any new articles. For this review, articles were selected and read by two reviewers (RRG, DG). Disagreements during this phase were resolved by consulting a third reviewer.

## Extracted data and assessment of bias

The data extracted were: ERP study objective, task type (or ERP paradigm), EEG system used, study population data, exercise characteristics (duration, intensity, pedaling cadence), ERP and behavioral results. We chose to record the behavioral results (response time and error rate on the attentional task) because they can help in the analysis of the impact of cycling on ERPs. The studies mainly focused on a comparison of ERP amplitude or latency between pedaling and non-pedaling conditions. However, some studies focused on other comparisons, *e.g.*, several exercise intensities (*Bullock, Cecotti & Giesbrecht, 2015*; *Olson et al., 2016*; *Dodwell et al., 2021*) or pedaling cadences (*Akaiwa et al., 2022*). Finally, two others took into account the effect of exercise duration (*Grego et al., 2004*; *Olson et al., 2016*).

Aerobic exercise intensity is an essential parameter to explore because it can influence the amplitude and latency components of ERPs after acute exercise (*Gusatovic et al., 2022*). In this review, we attempted to assess the impact of intensity.

# RESULTS

## Search results

The equation produced around 3,500 results, according to the search engine. Only the first 1,000 results (first 100 pages) were accessible. However, all the articles in this study were found in the first 500 results. Of the 1,000 results available on Google Scholar, 72 were selected by title. The abstracts were then read and 15 were eliminated because the measurements had been taken after; and not during exercise. Others did not concern ERP measurement, or the exercise used was not cycling. Nineteen abstracts were selected. Finally, full-text articles were explored. E-mails were sent to the authors of two studies to find out whether ERPs were actually measured during cycling. In the end, a total of 17

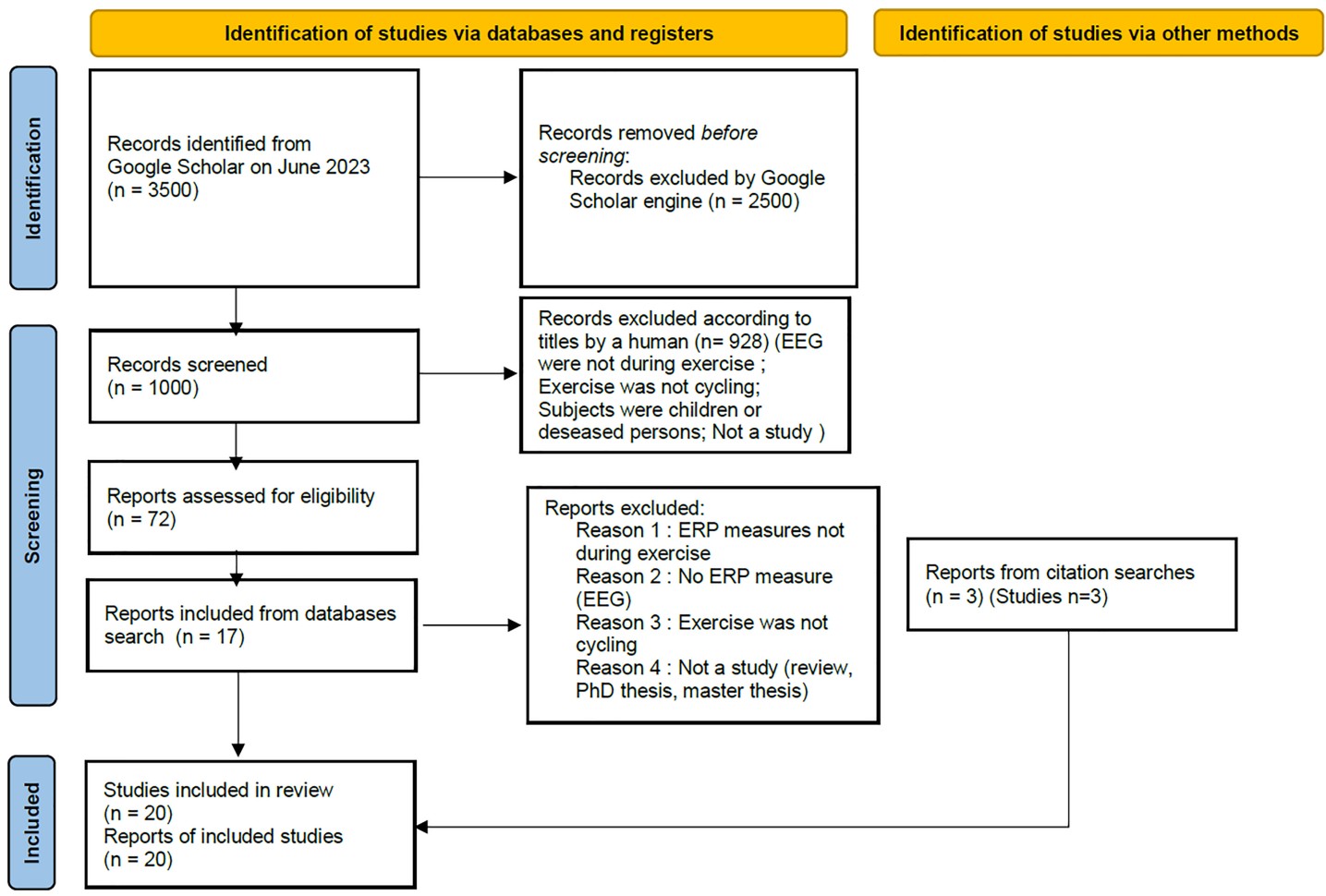

**Figure 1** **PRISMA 2020 flow diagram for new systematic reviews which included searches of databases, registers and other sources.** The pattern has been modified compared to the original one to make it more readable and adapted to our research process.

studies were retained. The remaining three studies included in the review (*Zink et al., 2016*; *Scanlon et al., 2017*; *Robles et al., 2022*) were not found during the search, as they do not contain the exact phrase "event-related potentials" in their full text. They were identified through previous searches with other equations or from references in the selected articles. We used the "PRISMA 2020 flow diagram for new systematic review which included searches of databases, registers and other sources" model (*Page et al., 2020*) as shown shown in Fig. 1.

## Study objectives

### Effects of cycling (intensity, cadence, duration)

Among the aims of the studies, the most common objective was to evaluate the effect of cycling on cognitive abilities and certain attention-related ERPs. Most studies compared the amplitude and latency of ERPs induced by an attentional task between a resting condition and a cycling condition. The cycling condition could be performed at low or moderate intensity, with three studies including two levels of intensity

(*Bullock, Cecotti & Giesbrecht, 2015*; *Olson et al., 2016*; *Dodwell et al., 2021*). Two studies assessed the effect of exercise duration on ERPs (*Olson et al., 2016*; *Grego et al., 2004*). Finally, one study (*Akaiwa et al., 2022*) compared several cycling conditions at the same intensity with three cadences, *i.e.*, at an optimal pedaling frequency chosen by the participants, 30% slower and 30% faster.

In some cases, the investigation of the effect of cycling on ERPs compared to a resting condition (without cycling) was not part of the objectives. For three studies (*Schmidt-Kassow et al., 2013*; *Conradi et al., 2016*; *Schmidt-Kassow, Thöne & Kaiser, 2019*), the aim was to test the effect on attention of synchronizing pacing with periodic auditory stimuli. In the study by *Grego et al. (2004)* the aim was to assess the effect of the duration of a long bout of athletic cycling on physiology and attention. In *Scanlon et al. (2020)* and *Robles et al. (2022)* the aim was to compare the effect of more or less noisy or hectic environments during outdoor cycling. The studies' objectives regarding ERP are briefly summarized in Table 1.

### Intention to approximate natural conditions

Nearly half the studies in this review aimed to approximate ERP measurements under "natural conditions", for example cycling in an outdoor environment. In *Killane, Browett & Reilly (2013)*, the aim was to test the feasibility of collecting EEG measurements of attention under so-called "ecological" conditions (cycle ergometer and treadmill). *Vogt et al. (2015)* evaluated the effect of exercise on a cycle ergometer with or without virtual movement on a virtual reality road. In *Zink et al. (2016)*, the experiment was carried out in an outdoor environment on a bicycle. The next five studies in this category all come from the same research team. In *Scanlon et al. (2017)*, the feasibility of ERP collection during sub-aerobic cycling was tested. In *Kuziek et al. (2018)*, the aim was to demonstrate that a minicomputer (the Latte Panda) could be used to accurately collect EEG data and perform EEG experiments in a portable manner. In *Scanlon et al. (2019)*, the aim was to test the feasibility of collecting EEG measures while cycling in a park. Even more advanced, *Scanlon et al. (2020)* investigated the effect of environmental noise under natural conditions on auditory ERPs while cycling. Finally, in *Robles et al. (2022)*, the aim was to study how a change in urban environment (more or less noisy) caused changes in attentional marker ERPs.

## Study features
### Participants

The number of participants was less than 17 in eight articles, and more than 19 for the other 12. The subjects included in *Grego et al. (2004)* were all male cyclist-athletes. The other studies mixed both sexes. However, the sex of participants was not specified in *Robles et al. (2022)*. The study by *Killane, Browett & Reilly (2013)* did not give an average age. A total of 14 out of 20 studies had an average age between 20 and 25 years, and five between 25 and 40. In addition, only 7 out of 20 studies reported selecting only right-handed people for their study. Table 1 shows more detailed information about the study populations.

**Table 1 Studies characteristics (population, type of assessment about ERPs, ERP paradigm, pedalling mode).**

| References | What is assessed about ERPs | Number of subjects in analysis | Mean age of subjects | Hand laterality of subjects | ERP paradigm | Pedalling mode |
|---|---|---|---|---|---|---|
| *Yagi et al. (1999)* | ERP modulations due to exercise | 24 | 21 | Right-handed | Visual oddball, auditory oddball (in two different conditions) | Recumbent bicycle ergometer |
| *Grego et al. (2004)* | Effects of duration of a long cycling exercise on cognitive functions | 12 (only men) | 29 | NA | Auditory oddball | Recumbent bicycle ergometer |
| *Pontifex & Hillman (2007)* | ERP modulation due to exercise | 41 | 20 | Right-handed | Modified flanker | Cycling ergometer |
| *Killane, Browett & Reilly (2013)* | ERP modulation due to exercise | 7 | 22 to 32 (no average) | NA | Auditory oddball | Fixed cycling |
| *Schmidt-Kassow et al. (2013)* | Effect of periodicity of sounds on attention allocation while cycling | 14 | 24 | Right-handed | Auditory oddball | Cycling ergometer |
| *Vogt et al. (2015)* | ERP modulation due to cycling in virtual reality | 22 | 30 | NA | Mental arithmetic | Cycling ergometer |
| *Bullock, Cecotti & Giesbrecht (2015)* | ERP modulation due to exercise | 12 | 20 | NA | Visual oddball | Recumbent bicycle ergometer |
| *Torbeyns et al. (2016)* | Effect of pedaling at an office desk on cognitive performances | 23 | 35 | NA | Rosvold continuous performance test | Chair-pedaler |
| *Olson et al. (2016)* | ERP modulation due to exercise and duration of exercise | 27 | 20 | Right-handed | Modified flanker | Cycling ergometer |
| *Zink et al. (2016)* | ERP modulation during outside pedaling | 15 | 27 | NA | Auditory oddball | Bike |
| *Conradi et al. (2016)* | Comparison of effect on attentional allocation between synchronising pace on periodic sounds or the sounds synchronising to the pace while cycling | 18 | 21 | Right-handed | Auditory oddball | Cycling ergometer |
| *Scanlon et al. (2017)* | ERP modulation due to pedaling | 14 | 25 | NA | Auditory oddball | Bike |
| *Kuziek et al. (2018)* | Check the performance of a portable informatic device (Latte Panda) | 16 | 21 | Right-handed | Auditory oddball | Stationary bike |
| *Scanlon et al. (2019)* | Good feasability during outside pedaling | 12 | 23 | NA | Auditory oddball | Bike |
| *Schmidt-Kassow, Thöne & Kaiser (2019)* | Effect of periodicity of syllables on ERP while cycling | 20 | 24 | Right-handed | Auditory oddball | Cycling ergometer |
| *Scanlon et al. (2020)* | ERP modulation due to level of noise during outside pedaling | 10 | 23 | NA | Auditory oddball | Bike |
| *Akaiwa et al. (2022)* | Influence of pedaling speed on attentional resources | 25 | 23 | NA | Tactile oddball (small electrical currents) | Stationary bike |

*(Continued)*

| Table 1 (continued) | | | | | | |
|---|---|---|---|---|---|---|
| References | What is assessed about ERPs | Number of subjects in analysis | Mean age of subjects | Hand laterality of subjects | ERP paradigm | Pedalling mode |
| *Dodwell et al. (2021)* | ERP modulation due to exercise | 24 | 23 | Three left-handed | Additional-singleton search task | Recumbent stationary cycling ergometer |
| *Robles et al. (2022)* | ERP modulation due to level of noise during outside pedaling | 24 | 21 | NA | Auditory oddball | Bike |
| *Olson, Cleveland & Materia (2023)* | Effect of light-intensity aerobic exercise on cognitive functions | 27 | 23 | NA | Visual oddball | Recumbent stationary bike |

Some studies excluded subjects after recording for various reasons. *Olson et al. (2016)* excluded three subjects because more than 50% of their trials contained artifacts. *Kuziek et al. (2018)* excluded one subject due to "registration problems", *Scanlon et al. (2020)* excluded five subjects due to "technical problems". *Dodwell et al. (2021)* excluded eight; four because the error rate in the cognitive task was too high (>20%), and four because they failed to maintain the required physical effort. In *Schmidt-Kassow et al. (2013)*, *Conradi et al. (2016)*, *Schmidt-Kassow, Thöne & Kaiser (2019)*, between four and six subjects were excluded in each study due to technical problems, incomplete recordings or excessively artifactual EEG datasets. Since excluding participants from the analysis means excluding all recordings in any condition from these participants, no imbalance in the outcome data results from this process.

### Attentional task during cycling

In most studies in this review, the task used during cycling was an attentional task, an oddball task (15 out of 20 studies), a modified flanker task (*Pontifex & Hillman, 2007*; *Olson et al., 2016*), or another task that primarily engages attention (*Torbeyns et al., 2016*; *Dodwell et al., 2021*). These tasks present few physical constraints that would require subjects to move their heads, which would cause more artifacts in the EEG signal.

In the auditory oddball (used by 11 out of 15 studies), a series of short tones is presented at a certain pitch (*e.g.*, 1,000 Hz); and a random distribution of a "rare" tone, often one in five, represents the "target" to which the subject must respond by counting them, or be pressing a key as soon as they hear it. The visual oddball works on the same principle but with targets displayed on a screen. *Yagi et al. (1999)* used the two variants (visual and auditory) separately. In *Akaiwa et al. (2022)*, a "tactile" variant was used, in which subjects had to count the electrical stimuli generated by electrodes around their fingers.

Two studies (*Pontifex & Hillman, 2007*; *Olson et al., 2016*) used a modified "Flanker" task, in which the task was to press in the direction indicated by a target arrow surrounded by arrows pointing in the same direction (congruent condition) or in the opposite direction (incongruent condition).

**Table 2 Quantitative distribution of paradigms used in the studies to trigger ERPs.**

| Oddball paradigm | | | Flanker paradigm | Rosvold continuous performance | Additional-singleton paradigm | Mental arithmetic |
|---|---|---|---|---|---|---|
| Auditory oddball | Visual oddball | Tactile oddball (electric) | | | | |
| 12 | 3 | 1 | 2 | 1 | 1 | 1 |

Note:
One study used two paradigms separately: a visual and auditory oddball (*Yagi et al., 1999*).

The study by *Torbeyns et al. (2016)*, which examined the effect on cognition of adapted cycling in office work, used the Rosvold Continuous Performance Test to measure ERPs. The task involved pressing a button when the letter X appeared. It appears to be similar to an oddball task.

The study by *Dodwell et al. (2021)* used the "additional-singleton paradigm" and involved indicating the orientation of the grating depicted in the target singleton (yellow circle), present among a set of green circles around the fixation point. A distractor was sometimes present in the form of a red circle. The task was said to be lateralized and supposed to involve a specific type of attention. A lateralized ERP called PCN (posterior-contralateral negativity) was calculated (see section on observed ERPs).

The study by *Vogt et al. (2015)* used a mental arithmetic task to test the combined effect of cycling and virtual reality on cognitive performance. This is the only study in which the task was not attentional. It involved guessing the correct result of a calculation as quickly as possible between two proposed results. The calculations were given in random order and were of equal difficulty. Table 2 presents the distribution of ERP paradigms among studies.

## EEG signal measurement and analysis
### Electrodes and EEG systems
The P3 wave is well marked at the Pz electrode, which is why it is often used for analysis in studies. The Cz and CPz electrodes are also often used. In *Dodwell et al. (2021)*, the NCP ERP was calculated from the activity of two contralateral parieto-occipital electrodes as follows: $((PO8 - PO7 \text{ (left singleton)}) + (PO7 - PO8 \text{ (right singleton)})/2)$.

Half of the studies (10) used a portable EEG, and the electrodes could be of the active type. Portable EEGs are wireless and can be used with a laptop in a backpack while cycling, as in *Scanlon et al. (2020)*. The studies by Scanlon and by Robles used a portable EEG (Brain Products Active Wet electrodes actiCAP) with active electrodes and used 15 electrodes out of 120 in a maximum configuration. In the study by *Zink et al. (2016)*, a portable EEG with 24 wet electrodes was used. In *Vogt et al. (2015)*, a portable EEG (Brain Vision Recorder, Brain products, Bavaria, Germany) with 64 electrodes was used. The details of the EEG systems used are presented in Table 3.

### Analysis of evoked potentials
All studies except (*Dodwell et al., 2021*) measured the P3 wave, which reflects a high-level attentional process, *i.e.*, with cognitive preprocessing. The studies are based on the increase in the P3 amplitude during the appearance of rare stimuli ("targets") in the oddball

**Table 3 Information about EEG system used, data processing and methods against artefacts.**

| References | EEG system | References and ground | Sampling rate in Hz | Frequency filters | Peak detection method of ERPs, time windows and electrodes of analysis | Use of seperate EOG, EMG and algorithmic methods against artefacts |
|---|---|---|---|---|---|---|
| Yagi et al. (1999) | Neuroscan stim system | Reference: left mandibles. | 500 | Bandpass was DC-100 Hz with a 60-Hz notch. Bandpass 1–30 Hz | The P300 component was identified as the largest positive deflection within a 180- to 600-ms latency window, at Pz | EOG: referential electrode pairs |
| Grego et al. (2004) | NA | Not defined | 256 | Bandpass: 0.01–30 Hz | largest positive deflection within a 250- to 500-ms latency window at Pz, Cz et Fz separately | Rejection threshold: +−100 microvolt. |
| Pontifex & Hillman (2007) | Neuroscan quik-cap (32) | Data were referenced to averaged mastoids (A1, A2) with AFz serving as the ground electrode and impedance | 250 | DC to 70 Hz filter, and a 60 Hz notch filter | The N1 and N2 components were defined as the largest negative-going peaks occurring within a 50–150 ms and a 150–300 ms latency window, respectively. The P2 and P3 components were defined as the largest positive-going peaks occurring within a 150–300 ms and a 300–600 ms latency window, respectively. | EOG: below the left orbit and on the outer canthus of each eye rejection threshold: +−100 microv |
| Killane, Browett & Reilly (2013) | Biosemi data acquisition system | Reference: left mastoid | 512 | Filtering included high pass (1 Hz), notch (47–53 Hz) and low pass (95 Hz) filtering | Maximum value of average P3 peaks at Cz, CPz and Pz separately | One EOG and three EMG electrodes included in the Biosemi system |
| Schmidt-Kassow et al. (2013) | Falk minow Services, Munich, Germany (64) | Fz served as ground and electrodes were referenced on-line to an average reference (recordings were reference offline: averaged mastoids | 500 | Band pass of 0.3–20 Hz Filtering included high pass (1 Hz), notch (47–53 Hz) and low pass (95 Hz) | P3 amplitude was the max amplitude at Cz, CPz and Pz separately | EOG (eight infra- and supraorbitally mounted electrodes were recorded); EMG ICA (Fast ICA) rejection threshold: +−100 microv |
| Vogt et al. (2015) | *Brain vision recorder 1.20.0701, brain products GmbH (64) | Reference: FCz Ground: Afz | 500 | low-pass 0.5 Hz, high-pass >50 Hz and notch 50 Hz | Amplitude at the peak following stimulus onset, for frontal, central, parietal and occipital regions (N200 and P300) | Electrode PO9 used as EOG |

| References | EEG system | References and ground | Sampling rate in Hz | Frequency filters | Peak detection method of ERPs, time windows and electrodes of analysis | Use of seperate EOG, EMG and algorithmic methods against artefacts |
|---|---|---|---|---|---|---|
| Bullock, Cecotti & Giesbrecht (2015) | BioSemi active two system (32) | Ref/ground: right and left mastoids | 512 | Low- and high-pass filters at 0.01 Hz and 30 Hz | Mean amplitude of P3a and P3b was calculated from a window around the peak latency (between 391 ± 25 ms for P3a and 423 ± 25 ms for P3b), averaging over channels CP1, CP2, Pz, P3, P4, PO3, and PO4. P1 mean amplitude was calculated by averaging data from occipital and parieto-occipital channels (Oz, O1, O2, PO3 and PO4), finding the peak latency of the positive going component between 100 and 150 ms post stimulus onset and calculating mean amplitude ±10 ms around this latency. | EOG: the left and right canthi and above and below each eye rejection threshold: +−125 microvolt |
| Torbeyns et al. (2016) | *brain products (32) | Re-referenced to an average reference | 500 | High pass: 0.1 Hz, low pass: 45 Hz and notch: 50 Hz; | Fronto-central region including Fp1, Fp2, F4, Fz, F3, F7, F8, FC1 and FC2 were used to analyse the N200 and the N450 and the temporal-parietal region including Pz, P3, P4, P7, P8, PO9 and PO10 were used to analyze the P300 and the conflict SP. Mean amplitude within a 150–300 ms latency window for the N200, a 250–500 ms latency window for the P300, a 450–550 ms latency window for the N450 and a 600–800 ms latency window for the conflict SP | ICA and inverse ICA |
| Olson et al. (2016) | Geodesic sensor net and electrical geodesics (64) | Reference: Cz reference off-line: left and right mastoids | 250 | Low-pass frequency of 30 Hz and high-pass frequency of 0.1 Hz. | N2: Fz, FCz, and Cz; P3: Cz, CPz, and Pz mean amplitude within a 200–350 ms window for N2 and within 250–500 for P3 | No |
| Zink et al. (2016) | *SMARTING mobile EEG amplifier from mBrainTrain (24) | Electrode used: FP1, FP2, Fz, F7, F8, FC1, FC2, Cz, C3, C4, T7, T8, CPz, CP1, CP2, CP5, CP6, TP9, TP10, Pz, P3, P4, O1 and O2. Rereferenced offline to the mean of TP9 and TP10 | 500 | High-pass filtered at 1Hz, and-pass filtered (0.5–20)Hz | The maximum P300 was computed at Pz in a (200–600) ms interval | Extended infomax ICA was used to remove EOG removal of the EMG was achieved through BSS-CCA method |

(Continued)

| References | EEG system | References and ground | Sampling rate in Hz | Frequency filters | Peak detection method of ERPs, time windows and electrodes of analysis | Use of seperate EOG, EMG and algorithmic methods against artefacts |
|---|---|---|---|---|---|---|
| Conradi et al. (2016) | Falk minow services (45) | Ground: Fz electrodes were referenced on-line to an average reference (recordings were rereferenced to averaged mastoids off-line) | 1,000 | Bandpass filter (110-140 Hz) for muscles artefacts, low-pass filter of 20 Hz | "we selected 15 frontocentral electrode sites that matched the P300-related ROIs" Peak detection method and time windows not defined | EOG: two electrodes placed over and under the right eye, FastICA for EOG artefects ICA rejection threshold: + −100 microvolt |
| Scanlon et al. (2017) | *Active wet electrodes (BrainProducts actiCAP) (32) | Ground: Fpz two reference electrodes: clipped to the left and right ear | 500 | Bandpass with cutoffs of 0.1 and 30 Hz notch filter at 60 Hz | P3: Average value at electrode Pz between 300 and 430 ms. MMN/N2b: Average value at electrode Fz between 175 and 275 ms. | EOG: above and below the left eye, and 1 cm lateral from the outer canthus of each eye. rejection threshold: +−100 microvolt |
| Kuziek et al. (2018) | *Brain products active wet electrodes actiCAP (32) | Ground: Afz references: eletrodes on the left and right ear lobes | 500 | Online bandpass with cutoffs of 0.629 and 30 Hz notch filter at 60 Hz. | Fz for MMN (between 175–275 ms) and Pz for P3 (between 300–550 ms) peak detection method (max or average not defined) | EOG: above and below the left eye, and 1 cm lateral from the outer canthus of each eye. rejection threshold: +−1,000 microvolt, then 500 microvolt |
| Scanlon et al. (2019) | *Brain products active wet electrodes actiCAP (32) | Ground: Fpz two reference electrodes: clipped to the left and right ear | 500 | Bandpass with cutoffs of 0.1 and 30 Hz notch filter at 60 Hz | MMN/N2b: electrode Fz averaged over the 175–275 ms time window P3: electrode Pz averaged over the 300–430 ms time window | EOG: above and below the left eye, and 1 cm lateral from the outer canthus of each eye. rejection threshold: +−1,000 microvolt, then 500 microvolts |
| Schmidt-Kassow, Thöne & Kaiser (2019) | Falk minow services (64) | Fz served as ground referenced to an average reference re-referenced to linked mastoids | 1,000 | Bandpass-filtered between 0.1 and 50 Hz First, to identify muscle artifacts, a band-pass filter (110–140 Hz) was applied low-pass filter of 20 Hz | P300 amplitude: average in the time-window 350–700, across the regions of interest. 11 electrodes were defined as constituting the central and parietal region of interest (including Pz) | Two electrodes placed above and below the right eye were used to control for eye movements The FastICA algorithm implemented in the FieldTrip toolbox was used to identify eye movement artifacts. The data underwent an independent component analysis (ICA). Removal of trials with amplitude differences between adjacent time points of more than 50 microvolt |

| References | EEG system | References and ground | Sampling rate in Hz | Frequency filters | Peak detection method of ERPs, time windows and electrodes of analysis | Use of seperate EOG, EMG and algorithmic methods against artefacts |
|---|---|---|---|---|---|---|
| Scanlon et al. (2020) | *Brain products active wet electrodes actiCAP (32) | Ground: Fpz two reference electrodes: clipped to the left and right ear | 500 | Bandpass with cutoffs of 0.1 and 30 Hz notch filter at 60 Hz | Mean amplitude within the time window 100–175 ms on Fz and Pz for the N1, 175–275 ms on Fz and Pz for the P2, 334–434 ms on Pz for the P3 | EOG: above and below the left eye, and 1 cm lateral from the outer canthus of each eye. rejection threshold: +−1,000 microvolt, then 200 microvolts |
| Akaiwa et al. (2022) | Neuropack system (?) | Referenced to linked earlobes (A1 A2) | 1,000 | Band-pass lter 0.1–300 Hz low-pass filtering at 100 Hz | P300 amplitudes were measured from baseline to peak, between 250–500 ms at Cz, Pz and Fz | EOG was recorded from the right suborbital region EMG mounted over the right vastus medialis (VM) and the short head of biceps femoris (BF) rejection threshold: +−200 microvolt |
| Dodwell et al. (2021) | *Brain products ActiCAP (64) | FCz serving as the online reference re-referencing all EEG signals to the 64-channel common average | 1,000 | 1 Hz high-pass and 50 Hz notch IIR filter 30-Hz low-pass IIR filter | event-related lateralization: the PCN (posterior-contralateral negativity) was observed in the grand-averaged waveform of lateralized target trials (lateral-target/midline distractor, lateral-target/no-distractor) (230–290 ms) | EOG: inferior orbit of the left eye ICA Infomax rejection threshold: +−60 microvolt |
| Robles et al. (2022) | Brain products active wet electrodes actiCAP (32) | Ground: Afz referenced to the left mastoid, and re-referenced to the arithmetically derived average of the left and right mastoids offline | 1,000 | Filtered online between 0.1 and 30 Hz notch filter at 60 Hz | ERPs: grand-average over 100 ms around the peak, N1: 118–228 ms maximal at FZ P2: 218–318 ms at Fz and Pz P3: 355–505 ms maximal at Pz | EOG electrodes were affixed vertically above and below the left eye and affixed horizontally 1 cm lateral from the outer canthus of each eye. rejection threshold: +−500 microvolt |
| Olson, Cleveland & Materia (2023) | *Brain vision actiCap, brain product (32) | Reference: Cz re-referenced to the left and right mastoids | 500 | Low-pass frequency of 30 Hz and high-pass frequency of 0.1 Hz | P300: Amplitude was measured as the mean amplitude within an *a priori* time window of 300–700 ms post-stimulus onset, across centro-parietal electrode sites (Cz, CP1, CP2, Pz) | EOG: above and below the left eye and approximately 1 cm lateral to the outer canthus of each eye ICA trials with a difference of 100 µV between minimum and maximum values in that trial were removed |

**Note:**
* The asterisk indicates that a mobile EEG system was used.
paradigm. In theory, this occurs 300 milliseconds after stimulus onset. Furthermore, the study by *Bullock, Cecotti & Giesbrecht (2015)* distinguished two components of the P3 wave in their analysis, called P3a and P3b, and assumed to be related to different and successive cognitive processes. MMN (Mismatch Negativity) is a component of the event-related response to an odd stimulus. This negative deviation occurs between 175 and 275 ms after the stimulus. This signal response appears when a rare deviant stimulus (as in the oddball paradigm) occurs among a frequent standard stimulus. MMN was calculated in the following three studies (*Kuziek et al., 2018*; *Scanlon et al., 2017*, *2019*). The study by *Dodwell et al. (2021)* measured a lateralized ERP, named PCN for "posterior-controlateral negativity", which is related to lateralized visual attention, and to the side on which the target stimulus appears. It was calculated from the activity of two diametrically opposed parieto-occipital electrodes. Other ERPs measured in the studies in this review include the N2, N1 and P2 waves, which are related to lower-level attentional processes, such as attentional orienting or background noise filtering.

### Methods to combat artifacts

There are several ways of limiting EEG artifacts.

Firstly, the experimental environment can be adapted. A room "shielded" from external electromagnetic sources, and minimizing electromagnetic sources inside the room, can help to reduce electromagnetic disturbances on the EEG, as in *Scanlon et al. (2017*, *2019)* and *Kuziek et al. (2018)*.

In addition, since cycling involves lateral head movements and artifacts, a cycle ergometer can be used on which you can lean strongly forward, as with road racing bikes, as in *Bullock, Cecotti & Giesbrecht (2015)*. Another solution is to use the seated position on a chair or a cycling system adapted to the seated position, as in *Torbeyns et al. (2016)*. The mode used in each study is detailed in Table 1. In addition, motion-adapted EEG equipment can also be used to minimize artifact, such as a wireless mobile EEG. A "breathable" mesh cap and an air-conditioned room can also limit artifacts due to perspiration (*Bullock, Cecotti & Giesbrecht, 2015*). Finally, participants can be encouraged to behave in a way that limits their head movements. They can be asked to stare at a cross on the screen between trials. Furthermore, low exercise intensity results in fewer EEG artifacts than high intensity exercise, in general, due to reduced head movements. Finally, *Grego et al. (2004)* asked subjects to close their eyes during recording, in order to limit artifacts associated with blinking or eye movement.

Secondly, the studies used a variety of post-recording methods to limit or deal with artifacts, in addition to the systematic use of frequency filters. High-pass, low-pass and band-pass frequency filters are commonly used in EEG. Most studies also used semi-automatic rejection of parts of the signal that were above an amplitude threshold value. A frequently used threshold value is 100 microvolts (see Table 3). Electro-oculogram (EOG) sensors have also been used to deal with ocular artifacts (blinks, saccades). They were placed around the eyes, above or below the left eye, to detect ocular saccades and blinks. EEG electrodes placed close to the eyes can make it possible to treat ocular artifacts without the need for EOG sensors. More advanced methods are less frequently used. A

regressive method of correcting ocular artifacts was used in five studies by the same research group, namely Scanlon, Kuziek and Robles. The "independent component analysis" (ICA) algorithmic method was used in some studies to reduce artifacts in the EEG recordings. A method to counter muscle artifacts called BSS-CCA ("blind source separation-canonical correlation analysis") was used in the study by *Zink et al. (2016)*. In *Akaiwa et al. (2022)*; electromyographic (EMG) sensors on the leg (right vastus medialis) and arm (biceps femoris) were used to calculate a correlation between EMG activity and the P3 EEG wave. However, no significant correlation was found.

## Conditions of use
### Duration
The studies by *Grego et al. (2004)* and *Olson et al. (2016)* had exercise durations of 180 min (single condition) and 30 min per condition respectively. These two studies had durations long enough to assess the specific effect of exercise duration. The studies by *Bullock, Cecotti & Giesbrecht (2015)*, *Dodwell et al. (2021)*, *Olson, Cleveland & Materia (2023)* had durations of 20 to 25 min per condition, but without the objective of testing the effect of duration. The duration of the conditions was not always mentioned and could be less than 12 min of cycling. It often consisted of several blocks of more than 100 tones in the oddball task, and the effect of duration was not included in the study objectives.

### Intensity
Several methods were used to determine the value of power (intensity). On the one hand, four studies based the power value on the results of an exercise test (*Grego et al., 2004*; *Pontifex & Hillman, 2007*; *Olson et al., 2016*; *Torbeyns et al., 2016*). On the other hand, in *Dodwell et al. (2021)* and *Olson, Cleveland & Materia (2023)* intensity was determined from a theoretical HRmax and a reserve HR. The formulae used by *Dodwell et al. (2021)* were as follows: HR_reserve = HR_max − HR_rest with HR_max = 208 − (0.7 × age). In *Olson, Cleveland & Materia (2023)* intensity was calculated as HRmax = 220 − age. However, most studies (*Killane, Browett & Reilly, 2013*; *Vogt et al., 2015*; *Zink et al., 2016*; *Scanlon et al., 2017*; *Kuziek et al., 2018*; *Scanlon et al., 2019*, *2020*; *Akaiwa et al., 2022*; *Robles et al., 2022*) did not give a precise value for intensity but rather, simply indicated "subaerobic", "low" or "moderate" intensity to qualify cycling intensity, as indicated in Table 4 detailing the exercise conditions in the included studies.

Exercise intensity varied across studies, but was most often mild, as the impact of intensity was not necessarily investigated. The study by *Grego et al. (2004)* consisted of constant exercise at 66% of the VO2 max for 3 h. In the study by *Yagi et al. (1999)*, intensity was adapted so that the heart rate (HR) was between 130 and 150 bpm. In *Pontifex & Hillman (2007)* the intensity corresponded to 60% of the HRmax on the basis of an exercise test. In *Olson, Cleveland & Materia (2023)*, intensity was defined as 60% of the theoretical HRmax (220-age). In *Torbeyns et al. (2016)*, subjects cycled at 30% of their estimated maximum power. In some studies (*Schmidt-Kassow et al., 2013*; *Conradi et al., 2016*; *Schmidt-Kassow, Thöne & Kaiser, 2019*), intensity was set at the same value (namely 50 W) for all subjects, corresponding to a light exercise intensity.

**Table 4 Exercise conditions (intensities and cadence).**

| References | Intensity of cycling | Other cycling intensity | Cadence in rpm |
|---|---|---|---|
| *Yagi et al. (1999)* | 130–150 bpm (HR) | No | 60 |
| *Grego et al. (2004)* | 66% VO$_2$max | No | Free pedaling rate |
| *Pontifex & Hillman (2007)* | 60% HRmax | No | Steady pace |
| *Killane, Browett & Reilly (2013)* | Light intensity | No | Self-paced |
| *Schmidt-Kassow et al. (2013)* | 50 watt | No | 60 |
| *Vogt et al. (2015)* | Moderate | No | Self-paced |
| *Bullock, Cecotti & Giesbrecht (2015)* | 40 watt | RPE12–14 (70 to 120 W according to participants' RPE) | 50 |
| *Torbeyns et al. (2016)* | 30% of max power | No | 80 |
| *Olson et al. (2016)* | 40% VO$_2$max | 60% VO$_2$max | Steady pace 50–75 rpm |
| *Zink et al. (2016)* | Biking freely | No | Slowly |
| *Conradi et al. (2016)* | 50 watt | No | 60 or self-paced (two different conditions) |
| *Scanlon et al. (2017)* | Sub-aerobic | No | Slowly |
| *Kuziek et al. (2018)* | Low-intensity | No | Evenly and constantly |
| *Scanlon et al. (2019)* | Sub-aerobic | No | Slowly |
| *Schmidt-Kassow, Thöne & Kaiser (2019)* | 50 watt | No | 60 |
| *Scanlon et al. (2020)* | Sub-aerobic | No | Slowly |
| *Akaiwa et al. (2022)* | NA (but very low scores of exertion) | No | Self-paced + 30% slower + 30% faster |
| *Dodwell et al. (2021)* | 40–50% reserve HR | 60–70% reserve HR | Self-paced |
| *Robles et al. (2022)* | Sub-aerobic | No | Slowly |
| *Olson, Cleveland & Materia (2023)* | 60% of HRmax (220-age) | No | Self-selected |

**Note:**
NA, not available; RPE, Rating of perceived exertion; HR, heart rate; HRmax, theoretical maximum heart rate.

Three studies (*Bullock, Cecotti & Giesbrecht, 2015*; *Olson et al., 2016*; *Dodwell et al., 2021*) each included a lower-intensity condition and a higher-intensity condition, with different methods for determining these intensities. *Bullock, Cecotti & Giesbrecht (2015)* called the cycling condition at 40 W "low-intensity exercise", and "high-intensity" was then determined by obtaining an effort score between 12 and 14 on the Borg scale. In *Olson et al. (2016)*, the low-intensity condition corresponded to 40% of the VO$_2$ max calculated using an exercise test, and the higher-intensity condition to 60% of VO$_2$ max. In *Dodwell et al. (2021)*, the lower intensity condition (called "moderate" in that study) corresponded to 40–50% of HR reserve (HRR) estimated from the formula: HR_max = 208 − (0.7 × age) then HR_reserve = HR_max − HR_rest, while the higher intensity condition corresponded to 60–70% of HR_reserve (called "vigorous" exercise in that study).

### Pedaling frequency or cadence

Cadence or pedaling frequency was sometimes imposed at a specific value, as in *Bullock, Cecotti & Giesbrecht (2015)*, *Yagi et al. (1999)* and *Torbeyns et al. (2016)* with imposed cadences of 50, 60 and 80 rpm, respectively. In three studies (*Schmidt-Kassow et al., 2013*;

*Conradi et al., 2016*; *Schmidt-Kassow, Thöne & Kaiser, 2019*), the cadence used was 60 Hz. *Dodwell et al. (2021)* mentioned the 70–80 rpm window, *Olson et al. (2016)* left the choice at the participant's discretion between 50 and 75 rpm, but the cadence had to be kept constant.

Other studies allowed any cadence (*Grego et al., 2004*; *Vogt et al., 2015*; *Akaiwa et al. 2022*) but with the intention of obtaining a constant cadence. Some studies did not specify a value but required a slow, steady cadence, as in *Scanlon et al. (2017*, *2019*, *2020)*, *Robles et al. (2022)*. No information on cadence was mentioned in the studies by *Killane, Browett & Reilly (2013)*, *Zink et al. (2016)*.

## ASSESSMENT OF RISK OF BIAS IN STUDIES USING THE COCHRANE RISK OF BIAS ASSESSMENT TOOL 2

In this section, we present the results of our analysis of the risks of bias using the Cochrane risk of bias tool 2. The results are summarized in Table 5. The sources of bias considered in this tool are:

- Bias in missing data for results.
- Measurement bias.
- Randomization bias.
- Bias in deviations from planned intervention.
- Bias in the selection of reported results.

Measuring the EEG signal during movement can be tricky because of movements of the head or part of the head, which create artifacts. For each study, it is plausible that larger parts of the EEG signal were suppressed in the cycling condition compared with the EEG signal in the resting condition. This is a potential source of bias in terms of missing data on the results. There may therefore be an imbalance in the outcome data. However, the studies did not clearly report this problem. It is possible that they did not have greater difficulty processing EEG data in the cycling condition than in the resting condition. Some studies included brief assessments of artifacts between the conditions, including RMS data noise (*Scanlon et al., 2017*, *2019*). They show that the cycling condition induces more data noise and may require a greater number of trials to achieve the same statistical power than the non-cycling condition.

All studies were categorized as "somewhat concerning" with regard to this form of bias.

According to the Cochrane tool, bias in outcome measurement concerns the possible influence of knowledge of the intervention received. However, this problem does not arise in the studies in this review. All studies were classified as "low risk of bias" on this point.

Randomization in the studies could not be concealed, since it is not possible to conceal the fact of cycling or not. Most of the studies in this review had a crossover randomization design, with the exception of *Killane, Browett & Reilly (2013)*, in which no randomization was reported, and *Grego et al. (2004)*, where randomization was impossible due to the experimental design. All but two studies were classified as "low risk of bias". *Killane, Browett & Reilly (2013)* and *Grego et al. (2004)* were classified as "high risk".

**Table 5 Assessment of risk of bias according to the Cochrane risk of bias tool 2.**

| References | Randomisation | Missing outcome data | Measurement of the outcome | Deviations from intended intervention | Selection of the reported results | Total of risk of bias |
|---|---|---|---|---|---|---|
| *Yagi et al. (1999)* | Low risk of bias | Some concerns | Low risk of bias | Low risk of bias | Low risk of bias | Moderate |
| *Grego et al. (2004)* | High risk of bias | Some concerns | Low risk of bias | Low risk of bias | Low risk of bias | High |
| *Pontifex & Hillman (2007)* | Low risk of bias | Some concerns | Low risk of bias | Low risk of bias | Low risk of bias | Moderate |
| *Killane, Browett & Reilly (2013)* | High risk of bias | Some concerns | Low risk of bias | Low risk of bias | Low risk of bias | High |
| *Schmidt-Kassow et al. (2013)* | Low risk of bias | Some concerns | Low risk of bias | Low risk of bias | Low risk of bias | Moderate |
| *Vogt et al. (2015)* | Low risk of bias | Some concerns | Low risk of bias | Low risk of bias | Low risk of bias | Moderate |
| *Bullock, Cecotti & Giesbrecht (2015)* | Low risk of bias | Some concerns | Low risk of bias | Low risk of bias | Low risk of bias | Moderate |
| *Torbeyns et al. (2016)* | Low risk of bias | Some concerns | Low risk of bias | Low risk of bias | Low risk of bias | Moderate |
| *Olson et al. (2016)* | Low risk of bias | Some concerns | Low risk of bias | Low risk of bias | Low risk of bias | Moderate |
| *Zink et al. (2016)* | Low risk of bias | Some concerns | Low risk of bias | Low risk of bias | Low risk of bias | Moderate |
| *Conradi et al. (2016)* | Low risk of bias | Some concerns | Low risk of bias | Low risk of bias | Low risk of bias | Moderate |
| *Scanlon et al. (2017)* | Low risk of bias | Some concerns | Low risk of bias | Low risk of bias | Low risk of bias | Moderate |
| *Kuziek et al. (2018)* | Low risk of bias | Some concerns | Low risk of bias | Low risk of bias | Low risk of bias | Moderate |
| *Scanlon et al. (2019)* | Low risk of bias | Some concerns | Low risk of bias | Low risk of bias | Low risk of bias | Moderate |
| *Schmidt-Kassow, Thöne & Kaiser (2019)* | Low risk of bias | Some concerns | Low risk of bias | Low risk of bias | Low risk of bias | Moderate |
| *Scanlon et al. (2020)* | Low risk of bias | Some concerns | Low risk of bias | Low risk of bias | Low risk of bias | Moderate |
| *Akaiwa et al. (2022)* | Low risk of bias | Some concerns | Low risk of bias | Low risk of bias | Low risk of bias | Moderate |
| *Dodwell et al. (2021)* | Low risk of bias | Some concerns | Low risk of bias | Low risk of bias | Low risk of bias | Moderate |
| *Robles et al. (2022)* | Low risk of bias | Some concerns | Low risk of bias | Low risk of bias | Low risk of bias | Moderate |
| *Olson, Cleveland & Materia (2023)* | Low risk of bias | Some concerns | Low risk of bias | Low risk of bias | Low risk of bias | Moderate |

No concerns were noted regarding deviations from the planned intervention because the intervention was the same for all subjects in each study. In *Dodwell et al. (2021)*, subjects who did not provide the required physical effort were excluded from the analysis. All studies were classified as "low risk" with regard to this form of bias.

The next bias mentioned in the Cochrane tool is the selection bias of the reported results. Some articles in this review reported no results regarding P3 wave latency (*Olson et al., 2016*; *Conradi et al., 2016*; *Scanlon et al., 2017*; *Kuziek et al., 2018*; *Scanlon et al., 2019*; *Schmidt-Kassow, Thöne & Kaiser, 2019*; *Scanlon et al., 2020*; *Robles et al., 2022*). However, it was implied (implicitly or explicitly) in their aims or methods sections that they would not analyze the difference in ERP latency. In fact, we did not observe any potential problem with the selection of results. All studies were therefore classified as "low risk" with regard to this form of bias.

**Table 6  Main study results for any ERP, and behavioral results (response time and error rate).**

| References | ERP amplitude | ERP latency | Behavioral results |
|---|---|---|---|
| Yagi et al. (1999) | P3 ↓ (in visual and in auditory oddballs) | P3 ↓ (in visual and in auditory oddballs) | Response time ↓ <br> Error rate ↑ |
| Grego et al. (2004) | P3 ↑ (between h 1 and h 2 of cycling) | P3 ↑ (in h 2 of cycling) | NA |
| Pontifex & Hillman (2007) | P3 ↑ <br> N1 ↑ <br> P2 ↑ <br> N2 ↑ | P3 ↑ <br><br><br> N2 ↑ | Response time ↔ <br><br><br> Error rate ↑ |
| Killane, Browett & Reilly (2013) | P3 ↔ | P3 ↔ | NA |
| Schmidt-Kassow et al. (2013) | P3 ↑ | P3 ↔ | NA |
| Vogt et al. (2015) | P3 ↔ <br> N2 ↔ | P3 ↔ | Response time ↔ <br> Error rate ↔ |
| Bullock, Cecotti & Giesbrecht (2015) | P3a and P3b ↔ <br> P1 ↑ (in lower intensity compared to no cycling) | P3a Latency ↓ <br> P1 ↓ (in lower intensity compared to no cycling) | Response time ↓ (in higher intensity compared to no cycling) <br> Error rate ↔ |
| Torbeyns et al. (2016) | P3 ↔ <br> N2 ↔ | P3 ↔ | Response time ↓ <br> Error rate ↔ |
| Olson et al. (2016) | P3 ↑ <br><br> N2 ↑ | NA | Response time ↓ (in higher intensity compared to no cycling) <br> Response time ↔ (in lower intensity compared to no cycling) <br> error rate ↑ |
| Zink et al. (2016) | P3 ↓ (in free biking condition compared to fixed biking and compared to no cycling) <br> N1 ↔ | P3 ↔ | Error rate ↑ |
| Conradi et al. (2016) | P3 ↓ (in passive synchronisation compared to no cycling) | NA | NA |
| Scanlon et al. (2017) | P3 ↔ | NA | NA |
| Kuziek et al. (2018) | P3 ↔ | NA | NA |
| Scanlon et al. (2019) | P3 ↓ <br> N1 ↑ <br> P2 ↓ (in outside environment compared to sitted inside condition) | NA | NA |
| Schmidt-Kassow, Thöne & Kaiser (2019) | P3 ↑ (in synchronized condition) | NA | NA |
| Scanlon et al. (2020) | P3 ↔ <br> N1 ↑ <br> P2 ↔ (noisy environment compared to calm environment in outside biking) | NA | Response time ↔ <br><br> Error rate ↔ |
| Akaiwa et al. (2022) | P3 ↓ | P3 ↔ | Error rate ↑ (in slow cycling compared to no cycling) |

 

| References | ERP amplitude | ERP latency | Behavioral results |
|---|---|---|---|
| *Dodwell et al. (2021)* | Posterior contralateral negativity (PCN) ↑ *(in no cycling and vigorous exercise compared to moderate exercise)* | PCN ↔ | Response time ↓ *(in higher intensity compared to no cycling)* |
| | | | Error rate ↔ |
| *Robles et al. (2022)* | P3 Amplitude ↔<br>N1 Amplitude ↑ | NA | Response time ↓ *(in heavy traffic condition compared to other conditions)* |
| | P2 Amplitude ↓ | | Error rate ↔ |
| *Olson, Cleveland & Materia (2023)* | P3 ↓ | P3 ↔ | Response time ↔ |
| | | | Error rate ↔ |

Notes:
↑ = increase compared to non-cycling condition.
↓ = decrease compared to non-cycling condition.
↔ = no significant difference between cycling and non-cycling conditions.
NA: not available in the study article.
In italics and brackets: specifications re the conditions in which the results were observed.

Ultimately, using the Cochrane tool, only two studies (*Killane, Browett & Reilly, 2013*) were selected. *Killane, Browett & Reilly (2013)* and *Grego et al. (2004)* were classified as being at high risk of bias, while the others were classified as raising concern. *Killane, Browett & Reilly (2013)* is a conference article and raises the most concerns regarding the quality of the study. There were only seven subjects, with no information on the average age, or the laterality of the subjects, and the behavioral results, exact intensity value and cadence were given. In *Grego et al. (2004)*, there were only male athletes and no randomization was applied.

## Study results

Results were reported for various types of ERPs (P3, P2, *etc.*) or involved comparisons (*e.g.*, between cycling and non-cycling, or another condition with cycling). Table 6 presents all the studies' results about ERPs and behaviours (response time and accuracy).

## Results based on a cycling compared to a resting condition

The results presented in this sub-section address the effect of cycling on ERPs and ERP characteristics, compared to the "resting" condition, *i.e.*, a condition in which the ERP paradigm was performed without cycling. *Dodwell et al. (2021)* is the only study that did not observe the P3 wave, but the authors reported a greater amplitude of the posterior PCN ERP in the presence of a distractor.

Results regarding the P3 wave:

Amplitude:

Four studies reported a greater amplitude of the P3 wave (*Schmidt-Kassow et al., 2013*; *Olson et al., 2016*; *Schmidt-Kassow, Thöne & Kaiser, 2019*) in the cycling condition compared to the resting condition.

Six studies reported the contrary, namely a lower amplitude of the P3 wave (*Zink et al., 2016*; *Conradi et al., 2016*; *Scanlon et al., 2019*; *Olson, Cleveland & Materia, 2023*; *Akaiwa et al., 2022*; *Yagi et al., 1999*). Specifically, for *Zink et al. (2016)* and *Scanlon et al. (2019)*,

this change was observed in a free cycling condition outdoors, on a bicycle. In *Conradi et al. (2016)* this result was obtained bearing in mind that in one cycling condition, the auditory stimuli were automatically synchronized to the subjects' spontaneous pedaling cadence.

In six studies (*Killane, Browett & Reilly, 2013*; *Vogt et al., 2015*; *Bullock, Cecotti & Giesbrecht, 2015*; *Torbeyns et al., 2016*; *Scanlon et al., 2017*; *Kuziek et al., 2018*), there was no significant difference in the amplitude of P3.

P3 latency time:

Two studies reported reduced latency in the cycling condition compared to the resting condition (*Yagi et al., 1999*; *Bullock, Cecotti & Giesbrecht, 2015*). In *Bullock, Cecotti & Giesbrecht (2015)*, the P3 wave was observed as a function of the P3a and P3b components, with P3a corresponding to the P3 wave for distractors (right-facing faces) and P3b corresponding to the P3 wave for targets (left-facing faces). A lower latency of P3a was observed.

One study reported longer latency (*Pontifex & Hillman, 2007*)

Five studies report no significant difference in latency (*Killane, Browett & Reilly, 2013*; *Schmidt-Kassow et al., 2013*; *Zink et al., 2016*; *Olson, Cleveland & Materia, 2023*; *Akaiwa et al., 2022*).

Other ERPs:

Some studies observed P1, N1, P2 or N2 waves, which are earlier than P3 and more closely linked to sensory processing. In *Bullock, Cecotti & Giesbrecht (2015)*, a greater amplitude of the P1 wave was reported. In *Zink et al. (2016)*, no change was reported for N1, while a larger amplitude was observed in *Scanlon et al. (2019)*, and a smaller amplitude in *Pontifex & Hillman (2007)*. In *Olson et al. (2016)*, a greater amplitude of the N2 wave was reported (for both intensities compared to rest). *Pontifex & Hillman (2007)* found a lower amplitude. In *Vogt et al. (2015)* and *Torbeyns et al. (2016)*, there was no change in N2. In *Scanlon et al. (2019)* and *Pontifex & Hillman (2007)*, a greater amplitude of P2 was reported.

The three studies (*Scanlon et al., 2017*, *2019*; *Kuziek et al., 2018*) that evaluated the MMN (mismatch negativity) response found no significant effect of exercise.

## Response time

Five studies reported a reduction in response time (*Yagi et al., 1999*; *Bullock, Cecotti & Giesbrecht, 2015*; *Olson et al., 2016*; *Dodwell et al., 2021*; *Robles et al., 2022*) of which two (*Olson et al., 2016*; *Bullock, Cecotti & Giesbrecht, 2015*) compared only the higher intensity condition to the resting condition.

Six studies (*Pontifex & Hillman, 2007*; *Vogt et al., 2015*; *Olson et al., 2016*; *Scanlon et al., 2020*; *Olson, Cleveland & Materia, 2023*; *Bullock, Cecotti & Giesbrecht, 2015*) reported no significant differences of which two (*Olson et al., 2016*; *Bullock, Cecotti & Giesbrecht, 2015*) compared only the lower intensity condition to the resting condition.

Five studies reported an increase in the error rate with low cycling intensity (*Yagi et al., 1999*; *Pontifex & Hillman, 2007*; *Olson et al., 2016*; *Zink et al., 2016*; *Akaiwa et al., 2022*).

In particular, in *Akaiwa et al. (2022)*, a loss of accuracy was observed for the slow pedaling cadence condition.

Six studies reported no significant difference (*Vogt et al., 2015*; *Bullock, Cecotti & Giesbrecht, 2015*; *Torbeyns et al., 2016*; *Akaiwa et al., 2022*; *Dodwell et al., 2021*; *Olson, Cleveland & Materia, 2023*).

## Comparison between two intensity levels

Three studies tested the effect of several levels of cycling intensity, comparing not only cycling conditions with a resting condition, as mentioned above, but also two cycling conditions of different intensities (*Bullock, Cecotti & Giesbrecht, 2015*; *Olson et al., 2016*; *Dodwell et al., 2021*).

In the two studies that observed the P3 wave (*Bullock, Cecotti & Giesbrecht, 2015*; *Olson, Cleveland & Materia, 2023*), no differences in amplitude or latency were reported between the two intensity levels. On the other hand, *Bullock, Cecotti & Giesbrecht (2015)* reported higher latency of the N1 wave in the high intensity exercise condition compared to the low intensity condition, and a tendency towards a lower amplitude. There was also no difference for the N2 wave in *Olson et al. (2016)*. In *Dodwell et al. (2021)*, the presence of a distractor reduced the amplitude of the posterior PCN wave at high or resting intensity, but not at low intensity.

With regard to response times, a significant decrease from low to high intensity was observed for *Bullock, Cecotti & Giesbrecht (2015)*, *Olson et al. (2016)* and only a trend for *Dodwell et al. (2021)*. There was no significant difference in the error rate between the two intensities across the three studies.

## Results for characteristics other than exercise intensity

We present below the results of comparisons other than *vs* the resting condition.

The effect of duration was tested in *Grego et al. (2004)* and *Olson et al. (2016)* by performing a series of oddball paradigms at several timepoints during moderately intense cycling (60% and 66% VO$_2$max respectively). In addition, the effect of different cycling frequencies was tested (*Akaiwa et al., 2022*). Finally, the studies by *Scanlon et al. (2020)*, *Robles et al. (2022)* tested the effect of a more or less noisy environment on attention-related ERPs during cycling.

## Effect of duration

The study by *Grego et al. (2004)* compared P3 ERPs before and after exercise as well as during exercise at different intervals (at 3, 36, 72, 108 and 144 min), but did not compare ERPs during cycling with a resting condition. The aim was to study the effects on athletes during a 3-h exercise session. An increase in P3 amplitude was observed between the first and third hour (measurements at 72 and 108 min). In addition, an increase in P3 latency was observed during the third hour (measurements at 108 and 144 min). The study by *Olson et al. (2016)* also tested a potential effect of exercise duration by performing series of ERPs at 5, 15 and 25 min of exercise. A decrease in the amplitude of the P3 wave was observed on each series.

### Effect of cadence

In *Akaiwa et al. (2022)*, results were presented for various cadences (optimal, 30% faster and 30% slower). On the Pz electrode, the amplitude of the P3 wave was lower in the slower and faster cycling conditions compared with the optimal cadence.

### Effect of synchronizing the bicycle with periodic sounds

The studies by *Schmidt-Kassow et al. (2013)*, *Schmidt-Kassow, Thöne & Kaiser (2019)* reported that synchronizing the cadence with the sound rhythms of the oddball task (at 60 Hz) resulted in a greater amplitude of the P3 than in the non-cycling condition. In *Conradi et al. (2016)*, sounds were synchronized with the subject's rhythm (passive synchronization, not active as in the other two studies) and this was linked to a lower amplitude of the P3 compared with the non-cycling condition.

### The effect of a noisy or agitated environment during cycling

In the studies by *Scanlon et al. (2020)* and *Robles et al. (2022)*, which compared several cycling conditions, in more or less calm environments and without a static condition, no difference in the amplitude of P3 was found. Conversely, they found an increase in the N1 wave when moving from a quiet environment (outdoors with little traffic) to a noisier one (close to road traffic). Modulation of the P2 wave was only observed in *Robles et al. (2022)* with a decrease in amplitude between low and intermediate traffic conditions.

## DISCUSSION

### Feasibility: it is possible to measure ERP while cycling

All the studies in this review show that it is possible to reliably measure ERPs during exercise. The diversity of exercise conditions in this review shows that it is possible to study ERPs under conditions that might reasonably be expected to be difficult, such as 3 h of moderately vigorous exercise (66% VO$_2$max) (*Grego et al., 2004*). The studies by Scanlon and Robles found that noise levels in the external environment had an effect on ERPs N1 and P2 during cycling. The lower amplitude of N1 in a noisier environment and the higher amplitude of P2 up to a noise threshold (not precisely determined) makes it possible to confirm hypotheses about the sensory processing functions of these ERPs and to show that it is possible to measure these changes reliably during outdoor cycling.

Half of the studies used portable EEGs with different electrode systems or types (active, wet, gel). It is also possible to use a portable EEG with a 64-electrode configuration, as in the study by *Vogt et al. (2015)*. The use of many electrodes is not necessary to measure and analyze the P3 wave or other ERPs, but it could enable the use of source localization methods. Although specific sensors have often been used to detect ocular artifacts, EEG electrodes close to the eyes would enable ocular artifacts to be detected and managed well. *Vogt et al. (2015)* used the PO9 EEG electrode to detect lateral eye movements.

As far as the methods to combat artifacts are concerned, some studies used algorithms such as ICA, but their use is generally limited to basic EEG processing methods (frequency filtering, amplitude rejection threshold). Moreover, studies of more intense exercise conditions do not appear to have used more advanced methods. According to a recent

review (*Sadiya, Alhanai & Ghassemi, 2021*), hybrid methods combining several methods such as ICA and deep learning have recently been created and could contribute to progress in this field.

## Results analysis

The two studies that used a Flanker task reported an increase in P3 wave amplitude (*Pontifex & Hillman, 2007*; *Olson et al., 2016*). However, more specifically, this result was the same for the lower (40% VO$_2$max) and higher (60% VO$_2$max) intensity conditions in *Olson et al. (2016)*. Perhaps the Flanker and oddball tasks are linked to a different effect of exercise on the P3 wave. However, a different effect of the Flanker task compared with the oddball task does not appear in the results of the review by *Gusatovic et al. (2022)*. Indeed, in the review by *Gusatovic et al. (2022)*, 7 out of 16 studies (44%) using the Flanker task reported an increase in amplitude compared with 5/10 studies using an oddball task (auditory or visual). There was no clear trend towards increased P3 amplitude in studies using the Flanker task.

The low amplitude of the P3 wave and the lack of a significant difference between exercise and rest may be linked to low-intensity exercise. Indeed, low exercise intensities may not be sufficient to induce neurophysiological adaptation. Exercise has to be sufficiently intense to have an effect on the P3 wave, as suggested by the theories of *Dietrich (2006)*, *Dietrich & Audiffren (2011)* and *Hebb (1955)* discussed below.

The reduction in response time and the increase in error rate seem to be linked to higher intensity. Response time never increases (in terms of significant difference) with exercise, but is sometimes reduced. Error rates are never lower during exercise, but sometimes higher than at rest. These results show that aerobic exercise can lead to a reduction in response time and an increase in error rate in an attentional task such as the oddball or Flanker tasks.

## Hypotheses or theories to explain results

All studies except (*Vogt et al., 2015*) used an attentional task. Among them, twelve used an auditory oddball task and only three used a visual oddball (see Table 2 for the total number using each paradigm). Attention is a limited resource that is important during cycling activity, particularly to manage external risks (*e.g.*, obstacles, road conditions). Observing the effect of exercise on the P3 wave could enhance our understanding of the effects of exercise on attention. All studies except (*Dodwell et al., 2021*) investigated the amplitude of the P3 wave. This ERP wave is involved in the attentional and cognitive processing of stimuli, and its amplitude is sensitive to the unexpected nature or relevance of stimuli. Some studies reported a decrease in the amplitude of the P3 wave during exercise compared with rest, while others reported an increase in P3 amplitude.

When a decrease in amplitude was observed, one hypothesis was the sharing of attentional resources in the brain, in order to perform several tasks simultaneously. This hypothesis has been proposed in studies such as *Yagi et al. (1999)*, *Scanlon et al. (2019)*, *Akaiwa et al., (2022)*. The hypofrontality transfer hypothesis (transient hypofrontality) of *Dietrich (2006)* goes in this direction. It is based on the idea that attention requires
metabolic resources whose availability is limited, and thus, motor and attentional processes operating in parallel are in competition. According to this theory, sufficiently strenuous exercise leads to a concentration of metabolic resources towards motor or exercise-relevant regions, to the detriment of regions less useful for exercise, such as the prefrontal regions. This could be the case for the parietal region for which the P3 wave was analyzed most frequently. According to this theory, executive control processes would benefit from simultaneous aerobic exercise up to a certain intensity or duration.

An increase in the amplitude of P3 in the exercise condition compared to the resting condition was observed in *Pontifex & Hillman (2007)* and *Olson et al. (2016)*. In both studies, the accuracy of subjects' responses to the Flanker task was poorer in the exercise condition than in the resting condition. According to *Pontifex & Hillman (2007)*, the concomitance of aerobic bicycle exercise and the attentional Flanker task results in increased recruitment of neural resources or decreased inhibition. Similarly, it was suggested in *Olson et al. (2016)* that there is a strengthening of attentional resources. However, this contradicts Dietrich's hypothesis. Furthermore, according to *Pontifex & Hillman (2007)*, the loss of response accuracy is linked to a loss of efficiency of neuroelectrical resources, which does not seem to fit well with the idea of "increased recruitment of neural resources".

In *Dodwell et al. (2021)*, other hypotheses were put forward. In the condition with a distractor in the attentional task, a greater amplitude of the ERP PCN was observed in the "moderate" cycling condition (40–50% HR reserve) compared with the resting condition, and a tendency towards a lower amplitude in the "intense" cycling condition (60–70% HR reserve) compared with the moderate condition. The hypothesis was that the facilitation of attentional or cognitive abilities follows an inverted U-shaped curve (*Hebb, 1955*), with optimal facilitation for low or moderate intensity exercise, and no facilitation for intense or non-existent exercise. However, these results were not replicated in the other two studies involving two conditions of different intensity and a rest condition (Bullock, Cecotti, Cecotti, Cecotti and Cecotti). *Bullock, Cecotti & Giesbrecht (2015)*, *Olson et al. (2016)*. This hypothesis therefore remains to be verified by further studies.

The ERPs P1, N1, P2 and N2 appear earlier than P3, and involve lower-level sensory processing. Considering the studies that compared results between exercise and resting, these ERPs were not treated systematically. For the lowest amplitude of N1 observed, the authors of *Pontifex & Hillman (2007)* suggested a deterioration in visual attention, while the increased amplitude of P2 suggested an increase in attentional selectivity, and the low amplitude of N2 suggested an attentional conflict.

However, these hypotheses are weak because they relate to results that have not been reproduced by other studies. Further studies are needed to assess the validity of each of these hypotheses or theories.

## Discussion of methods and limits of evidence

The studies in this review did not use same criteria to define exercise intensity. The validity of the definition of intensity in these studies could therefore be called into question. In *Bullock, Cecotti & Giesbrecht (2015)*, basing the determination of "high intensity" on the
assessment of perceived effort (using the Borg scale) is unreliable, especially for subjects who are not used to assessing their effort. In fact, the "high intensity" value varied from 70 W (which almost corresponds to a warm-up intensity) to 120 W, depending on the subjects. In *Torbeyns et al. (2016)*, the intensity was low and determined by an exercise test, but it is unclear whether this was a conventional exercise test, and the reliability is unknown. In addition, in several studies, target intensities were calculated only theoretically. As a result, it was not possible to classify the studies as low, moderate or vigorous intensity to assess the effect of intensity. Since exercise intensity and cadence were not always rigorously defined, there could be potential for bias due to different exercise conditions between subjects. According to *MacIntosh et al. (2021)*, the distinction between low and moderate-to-vigorous intensity should be determined by measuring a blood lactate threshold or ventilatory threshold. These physiological parameters are well linked to homeostatic change which, in theory, makes it possible to distinguish between low-intensity exercise and moderate- or vigorous-intensity exercise. Descriptions from American and Canadian institutions also suggest heart rate percentage values and other values that are associated with moderate or vigorous intensity levels.

Testing several intensities under the same conditions would be a reliable way of assessing the effect of intensity. However, only three studies did so (*Bullock, Cecotti & Giesbrecht, 2015*; *Olson et al., 2016*; *Dodwell et al., 2021*) and each had its own method of measurement and analysis, as well as different intensity criteria, thus making it difficult to compare their results. In addition, studies in which the exercise takes place in an outdoor environment (park, road) are exposed to environmental variability, such as weather parameters and traffic conditions (*Zink et al., 2016*; *Scanlon et al. 2019*, *2020*; *Robles et al., 2022*).

Exercise duration also varied across studies. In most cases, the exact duration of cycling was not mentioned. It consisted in performing several blocks of numerous short trials of the attentional task, sometimes with pauses between blocks. In some studies, the exercise condition lasted 20 min or more (*Bullock, Cecotti & Giesbrecht, 2015*; *Olson et al., 2016*; *Olson, Cleveland & Materia, 2023*; *Dodwell et al., 2021*), but this was probably not the case for most studies. However, the significance is not necessarily the same if ERP was measured at the start of cycling exercise, or at the end, or during exercise with recovery breaks.

Furthermore, the methods used to clean EEG data and quantify ERPs, such as amplitude, can affect the study results. For example, the amplitude of the P3 wave could be quantified as a peak value relative to baseline, or as the difference between the amplitude of rare tones in the auditory oddball and the amplitude of frequent tones comparing cycling and resting conditions as in *Olson, Cleveland & Materia (2023)*. However, we could not be sure of the exact method used to quantify ERPs in all studies. For example, in the methods section of *Torbeyns et al. (2016)*, it is stated that "The latency and amplitude of each ERP component were quantified using the mean amplitude and corresponding latency […]", but exactly what this means is unclear. This type of difference between studies may have less effect thanks to within-group models, but may nevertheless influence results based on the comparison between a cycling condition and a resting condition.

The study populations were predominantly mixed, and mostly aged between 20 and 25. However, the EEG signal can be influenced by a person's age (*Remijn et al., 2014*), which could have repercussions on measurement results. For example, one study found that older participants show a lower MMN amplitude than younger participants (*Gaeta et al., 1998*). Therefore, the results of the studies in this review cannot be generalized to all ages.

Finally, the laterality or handedness of the subjects was not systematically reported. Only seven studies out of 20 indicated that they had included right-handed subjects only. However, ERP components can be affected by handedness (*Remijn et al., 2014*), such as in motor preparation (*Schmitz et al., 2019*). It has also been shown that handedness can have an impact on the ERP components from a visual oddball task (*Eskikurt, Yücesir & İsoglu-Alkac, 2013*). For that reason, it may be problematic to mix right-handed, left-handed or ambidextrous subjects in the same study.

## Limits of this study

One difficulty in the search process for articles to include is that we cannot know whether ERPs were measured during cycling with titles such as "Effect of cycling on ERPs" because it may be related to EEG measurements after acute exercise only. We did not find a search equation that made it possible to select ERP measurements performed only during cycling without resulting in some of the included studies being overlooked.

This review included some studies that did not have a resting condition to compare ERPs with a cycling condition. This complicates the comparison and organization of results due to heterogeneity.

The Cochrane Risk of Bias Assessment Tool 2 was used to assess the risk of bias in the included studies. The categories in this tool concern bias within studies. They differ from the problem of methodological heterogeneity between studies. The tool comprises five categories of possible bias. It was found that some categories do not correspond well to the type of studies in this review, notably "deviations from planned intervention", "outcome measurement" and randomization bias. It seems that this tool is best suited to blinded studies of the efficacy of new treatments. It is possible that this tool was not applied appropriately, but there may be other instruments that are more suited to the type of studies included in this review. However, the "missing data" category found that the studies included did not consistently report a problem of imbalance in the amount of data analyzed between the exercise and resting conditions. We hypothesize that, due to movement, EEG data are more likely to be marred by artifacts in the cycling condition than in the resting condition. This could have an impact on the quantity and quality of data between the two conditions and bias the results. Overall, the Cochrane tool enabled us to conclude that there was a moderate risk for all studies due to the "missing data on results" category.

This review did not include studies that measured ERP during aerobic exercise other than cycling, such as rowing, walking or running on a treadmill. Including studies of other forms of exercise would broaden the range of results at the cost of greater heterogeneity in exercise conditions. The review by *Schmidt-Kassow & Kaiser (2023)* included the latter two types of exercise, but with a different objective from that of the present review.

Finally, this review did not provide detailed information on how to measure ERPs during cycling or how to deal effectively with artifacts.

## Perspectives

A recent study (*Schmidt-Kassow & Kaiser, 2023*) addressed the subject of behavioral and EEG measures during cycling, walking or running, with studies assessing various cognitive processes such as attention, inhibition, memory, vigilance and cognitive flexibility. They reported that behavioral studies assessed more conditions of intensity, duration and cognitive paradigms than EEG studies. However, most EEG studies (which are partly common to this review) assessed attentional abilities *via* an oddball task. According to the authors, in EEG studies, researchers were more concerned with demonstrating the reliability of ERP results in a motion condition. We speculate that this might be one reason why they used mainly an oddball task. We mentioned in the feasibility part of this discussion that ERPs can be measured in different cycling conditions. This fact gives confidence in the assessment of other aspects of cognition (other than attention) during aerobic exercise. This would make it possible to compare behavioral and EEG studies and assess the effect of aerobic exercise on various cognitive processes during exercise.

The studies in this review used ERP measurement during cycling for a variety of objectives. Notably, the studies by *Scanlon et al. (2019*, *2020)*, *Robles et al. (2022)* reported an impact of urban cycling on attention *via* ERPs. Their research may be linked to knowledge of the potential risks of this type of situation, since attention is a key ability in managing risks on the road.

ERPs during cycling may also be of interest in sports research, since (*Grego et al., 2004*) found changes in ERP amplitudes during a 3-h long race for cyclists. ERPs during cycling can also be linked to concentration during sports performance, and (*Schmidt-Kassow et al., 2013*; *Conradi et al., 2016*; *Schmidt-Kassow, Thöne & Kaiser, 2019*) found that actively synchronizing one's cycling cadence with a metronome had an impact on ERPs.

Some studies attempted to assess the impact of different exercise intensities on ERPs (*Olson et al., 2016*; *Bullock, Cecotti & Giesbrecht, 2015*; *Dodwell et al., 2021*), but further studies are needed to understand how intensity may impact components of ERPs such as amplitude and latency.

We could also imagine measuring ERP during cycling with objectives for overall health. Indeed, the ERP amplitude could be used as a marker of concentration or motivation, and could be applied in cardiac rehabilitation to assess motivation during a rehabilitation session.

Common definitions of intensity should be used to make it easier to compare results between studies. It might also be preferable to define exercise intensity using the same method. By taking a percentage of a theoretical heart rate reserve, as in *Dodwell et al. (2021)*, only the resting heart rate of the participants is required, and this is less demanding than doing an exercise test for each individual. The theoretical heart rate reserve method is easy and would be sufficient if studies also referred to official ratios to decide on intensity levels, as presented in Table 1 of *MacIntosh et al. (2021)*.

In addition, all exercise conditions should also be detailed in the articles. It is important to specify the cadence used during cycling sessions because it can influence intensity (even if the effect is small). Indeed, as shown in *Akaiwa et al. (2022)*, different cadences can lead to different behavioral and ERP results. The time at which measurements are taken is also important, since results do not have the same significance if recorded at the beginning or end of an exercise session, even if it is of light intensity.

In order to be able to extrapolate the results to the general population, researchers should include a wider range of ages, or carry out specific studies on older populations, such as cardiac rehabilitation patients, who could benefit from such research in the future, with the development of new technologies. It is also important to systematically mention the characteristics of the study population (number, age, gender and laterality), as these may have an impact on the results and be useful for future meta-analysis.

The paucity of studies on vigorous intensity and the variability of methods lead to considerable uncertainty as to the effect of intensity on attentional ERPs, including the most widely studied parameter, namely P3 amplitude. The P3 latency and other ERPs are reported less frequently. Additional studies are needed to include two cycling conditions of different intensity in order to better assess the immediate effect of cycling exercise on ERPs and attentional abilities.

## CONCLUSIONS

Technological advances over the last few decades have made it possible to perform EEG studies under motion conditions that were previously not possible due to signal artifacts. Our first aim was to examine feasibility. All the studies in this review showed that it was possible to measure ERPs during cycling. Our analysis of the risk of biases with the Cochrane tool 2 revealed that studies do not always report an imbalance in signal quality between the cycling and non-cycling conditions, which makes it difficult to assess whether there are more artifacts during exercise. The ERP most commonly investigated in this review was the P3 wave, which is involved in attentional and cognitive processes. The diversity of objectives and study results shows that there is a wide range of possible applications, such as for competitive sports or road safety. Our second aim was to assess the impact of cycling on ERPs during exercise, and the impact of intensity. This review shows that studies vary widely in their methods and their results. Consequently, it was impossible to draw any firm conclusions about the actual effect of cycling and intensity. Certain methodological aspects could be improved to overcome this difficulty. A precise description of all exercise conditions, including cadence, intensity and duration of exercise, would facilitate comparison of study results. In the same way, the method of calculating exercise intensity should be systematically based on reference criteria. Furthermore, systematic analysis of the latency and amplitude of several types of ERPs would provide a better understanding of the effect of cycling on these parameters. Finally, it would be interesting to use tasks other than attentional tasks, in order to study a wider range of potential effects of cycling exercise on ERPs. These improvements will make it possible to discuss theories of the effect of aerobic exercise on neurophysiological functions during aerobic exercise.

### Funding

The Conseil Régional de Bourgogne-Franche-Comté funded the doctoral grant of Rémi Renoud-Grappin. The funders had no role in study design, data collection and analysis, decision to publish, or preparation of the manuscript.

### Grant Disclosures

The following grant information was disclosed by the authors:
The Conseil Régional de Bourgogne-Franche-Comté.

### Competing Interests

The authors declare that they have no competing interests.

### Author Contributions

- Rémi Renoud-Grappin conceived and designed the experiments, performed the experiments, analyzed the data, prepared figures and/or tables, and approved the final draft.
- Lionel Pazart conceived and designed the experiments, authored or reviewed drafts of the article, and approved the final draft.
- Julie Giustiniani conceived and designed the experiments, authored or reviewed drafts of the article, and approved the final draft.
- Damien Gabriel conceived and designed the experiments, authored or reviewed drafts of the article, and approved the final draft.

### Data Availability

This is a systematic review/meta-analysis.

### Supplemental Information

Supplemental information for this article can be found online at http://dx.doi.org/10.7717/peerj.17448#supplemental-information.

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
