# Peer review of "State of the art and future directions for measuring event-related potentials during cycling exercise: a systematic review"

_PeerJ, doi:10.7717/peerj.17448_

## Round 0.1 · original submission · Major Revisions

As you can read from the reviewers' report, the reviewers have individuated several criticalities in the study. All the reviewers are unanimous in recommending a major revision of the manuscript.

Reviewer 1 ·

Basic reporting

a. I think one of the most important issues with the current version of the manuscript is grammar and spelling. Here are some notable examples of issues I found:
i. Introduction
1. I think you should combine the first two paragraphs, as the start if the second paragraph (line 57) is very disconnected from the rest of the paragraph.
ii. Materials and methods
1. Line 114 is very confusing/redundant, I recommend removing it or changing the wording.
2. Line 129 – “following” is used in the plural form when it is singular.
3. Line 149 – wrong form of analyses used.
b. The description of event-related potentials in lines 58-61 is slightly confusing and not a good representation of the underlying phenomenon. An ERP refers to the time-locked signal in response to some event, and not necessarily the changes in amplitude within that signal. The “spikes” are best described as ERP components.
c. The sentence spanning from lines 64-66 is confusing, although I think if you elaborated on what you mean by “constraints” it may be more clear.
d. What is the rationale for the point made in lines 68-70? Is there a research example of this? Similarly, in the next sentence, what is meant by “risks”? I am unsure if these components add to this manuscript in a meaningful way.
e. The research cited in line 73 does not seem to relate to the point being made, as the article referenced is mainly concerned with visual stimuli and cognitive experiments. I think a different research example would be more appropriate to make this point.
f. The sentence spanning lines 114-120 is a confusing, drawn-out description of how you described differences. I think it would be better to split this into multiple sentences for different measures.
g. I don’t think the line spanning lines 128-129 adds anything to your discussion of methods.
h. I would like to see more supporting citations on the P300 background throughout the methods, such as in the paragraph starting on line 293.

Experimental design

I have two issues with some of the descriptions of the design -
a. Can you include some rationale as to why you only included adults (line 99)?
b. Line123-124 I think you should include more rationale for this choice as it is central to your findings/methods.

Validity of the findings

Impact and novelty are appropriately assessed here. Results are accompanied with sufficient tables and other supporting information. Conclusions are well stated and clear.

Additional comments

One complaint with interpreting these findings is lack of discussion on ERP analysis in the studies mentioned. In the paragraph starting on line 328, authors discuss different pre-processing techniques used to clean EEG data, which may affect the resultant ERP component quantities. However, perhaps more importantly, the method used to quantify ERPs will have a great effect on how results are interpreted. If one study uses a max-peak method, while another uses a base-to-peak or mean measure, then it is harder to link conclusions from the two studies. Admittedly, this may have less of an effect in within-groups designs, but it should still be discussed here. This is an limitation that faces many research disciplines that hope to use ERPs in their experiments, and should be discussed as a relatively major factor here.

My main two issues with this manuscript are heavy grammar/spelling/phrasing fixes throughout and discussions related to ERP research practices. I recommend a rigorous editing process, as well as revisiting the sections of the manuscript related to ERPs and clarifying or elaborating the related discussions.

Reviewer 2 ·

Basic reporting

The review's language is professional, but it falls short in the accuracy of its technical descriptions, particularly regarding event-related potentials (ERPs). ERPs are not just "spikes of amplitude" but specific brain responses to stimuli, and this should be clarified and expanded upon in the introduction. Also, the introduction lacks essential citations, especially where claims about the history and application of EEG are made. These referencing gaps affect the review's credibility and fail to situate it within the existing literature properly. Furthermore, while logical, the introduction's structure is weakened by content inaccuracies and sparse referencing. Accurate definitions and sufficient references are critical for establishing the hypotheses and the knowledge gap the review aims to address. The introduction needs a thorough revision for precision and well-supported explanations to enhance the review's integrity and its contribution to scholarly discourse.

Suggested improvements:

1. Provide a detailed, accurate explanation of event-related potentials (ERPs) beyond just "spikes of amplitude," emphasizing their role and measurement in EEG studies.
2. Include essential references for claims about EEG's history and application, ensuring credibility and proper context within existing literature.
3. Revise the introduction to correct factual errors, aligning with current scientific understanding.
4. Articulate the review's research questions and the specific knowledge gaps it aims to address.

Experimental design

The methods described follow the PRISMA guidelines, which are appropriate for systematic reviews. However, the authors acknowledge that some items of PRISMA were not applicable due to the heterogeneous nature of the studies included. This is a reasonable approach, but the authors must ensure that the modifications made to their methodology because of the diverse studies are transparent and justified.

The research question is implied rather than explicitly stated. There should be a clearly defined research question or objective, which is the feasibility and impact of exercise on ERPs as measured during cycling exercise.

While the authors mention using the Cochrane risk of bias tool, they also note that some categories of bias were not applicable. There is a risk here that excluding these categories could undermine the comprehensiveness of the bias assessment. It’s important that the authors clearly justify these exclusions and discuss any potential impact on the validity of their findings.

The authors appear to have made broad assumptions about certain study parameters (e.g., the duration of cycling and intensity of exercise) when these were not specified, which could potentially skew the results and interpretations. This approach can seriously compromise the validity of the findings.
This section does not directly address conclusions; however, the methods outlined will directly impact the strength of the conclusions drawn. The reliance on assumptions and the potential for bias in article selection could limit the conclusions' relevance and accuracy.

Suggested improvements:

1. While following PRISMA guidelines is appropriate, any deviations due to the heterogeneity of included studies need clear justification and transparency.
2. Define the research question or objective clearly, focusing on the impact and feasibility of exercise on ERPs during cycling exercise.
3. If using the Cochrane risk of bias tool but omitting certain bias categories, provide clear reasons for these exclusions and discuss their potential impact on the findings' validity.
4. Avoid making broad assumptions about unstated study parameters like exercise duration and intensity, as these can affect the results' validity and skew interpretations.

Validity of the findings

The text does not specifically address the impact and novelty of the studies reviewed, which is a notable omission. There is a missed opportunity to critically evaluate how each study contributes new knowledge or replicates existing knowledge with an added benefit, as per the given criteria.

There is an explicit admission in the text that methodological aspects across studies could be improved, which casts doubt on the robustness and comparability of the underlying data. The failure to ensure uniformity and control in the methodological approaches significantly undermines the validity of any conclusions drawn.

The conclusions presented are cautious and highlight the inability to draw definitive conclusions regarding the effects of cycling on ERPs. This suggests that the studies may not adequately address the original research question or that the results are too inconsistent to support strong conclusions.

Suggested improvements:

1. Include a critical evaluation of each study's contribution to new knowledge or how it builds upon existing knowledge.
2. Acknowledge and address the methodological differences across studies, highlighting their impact on data robustness and comparability.
3. Clarify the conclusions, focusing on how the findings align with the research question and the implications of any inconsistencies in the results.

Additional comments

I recommend you do a deep dive into ERP literature.

Reviewer 3 ·

Basic reporting

The writing is often unclear and uses informal language, and the English could be improved. Some sections are not formatted into paragraphs. I believe the paper has a lot of potential but is currently lacking some attention to detail.

Experimental design

The paper tries to follow the standards of the Cochrane risk of bias tool 2. However after the section evaluating the studies, this tool is not mentioned very much in the discussion or conclusions. I would like to hear more about how the authors think these results should be interpretted for the reserach in the future.

Validity of the findings

As this is a systematic review, the findings seem quite valid since they all refer to other papers.

Additional comments

Summary:
This is a systematic review of Cycling-related EEG literature. It is well organized and has the potential to be a very useful tool for future researchers in this field. However it needs to be improved in the organization, quality of writing, as well as figures and tables. The writing is often unclear or uses informal language. The paper would also benefit from English editing.

Abstract:
‘Evoked-related potentials’
I think here you might mean to say Event-related potentials (also in the introduction)
Abstract methods: Please add a little more detail and/or clarity here. If the result is that ‘Twenty studies were selected’, please briefly explain how you selected them. If this is simply stating the selected studies, please include it in the methods section instead of results.

Selection criteria:
‘Studies applied to any desease were excluded.’ (line 99)
Do you mean to say studies were applied to a diseased population? Please rephrase.

Data extracted and assessment of bias:
‘It may be assumed it is the one that could have the most impact on cognition and thus ERPs.’ (line 123)
Please clarify what ‘it’ is.

‘it was assumed that duration of cycling were inferior to 11 minutes’ (line 138)
The term ‘inferior’ refers to the quality, when I think you mean here to say ‘less than’, which refers to the quantity of time used. Please reword.

Objectives of the studies:
Tasks used list (line 264-277):
This would be much easier to follow in a figure or table format.

‘Another ERP is the MMN’ (line 298)
This is not an ERP. This is a component of the ERP. Please reword. Also adjust this at line 305 and anywhere else it comes up in the manuscript.

Also I would consider changing the language of e.g. P300 or N200 to P3 and N2. The component itself can happen at varying times during ERP, depending on the stimulus, task, etc. Therefore using the shortened numbers refers to the number of peaks in the ERP. For example, the P3 is the third positive peak in the ERP signal. It can occur anywhere between 250-500ms or even later.

Tables 1 and 2:
You don’t need the studies’ citations in parentheses here.
Please keep abbreviations consistent (e.g. yrs or yo), and make sure they are clear to the reader.
Table 1 population column is too crowded, especially in the title. Please write in a more organized way. Replace ‘nb’ in reference to number, as this is not the common short form. Explain all short forms somewhere.

P. 17: Please cite the Cochrane risk of bias tool.

Line 429
Many of these studies did not have a hypothesis concerning e.g. latency of the P3, does that really make them biased? Every study has it’s own goals and aims, and papers would get very long and harder to read if we tried to include every possible measure. What exactly justifies the inclusion of P3 latency and reaction time as a measure of bias?

Line 472: Does ‘Under-ventilatory’ mean under the ventilatory threshold? If so, please write that. Otherwise adjust it to a common term so readers can understand.

Line 516: ‘It is quiet indoors for (Scanlon et al. 2019a) whereas it is outdoors for (Zink et al. 2016)’
Please rephase. It’s not clear what you are trying to say about the Zink study here.
Also please re-structure this whole section (and others like this) into an actual paragraphs

Line 607: ‘Effect of synchronising cycling with periodic sounds’
Why do you interpret the results in this paragraph but not for any of the other ones? Please keep it consistent.

Table 4: Please reorganize this table and try to make it clearer. Why do some lines have hyphens and some not? What does this symbol (↔) mean here? Try to organize it so that someone who was just reading this paper for the first time with no background could easily pull the meaning from it.

Line 639: ‘In (Gusatovic et al. 2022), 7 out of 16 (44%) measuring’
7 out of 16 what? Please clarify

Line 682: please rephrase, it’s not clear what you mean by ‘make it well’

Line 782: please rephrase this sentence: ‘But it doesn’t necessarily mean the same if measured were achieved at the beginning the cycling exercise or after a while’
What do you mean by ‘if measured were achieved’?

Line 798: ‘What’s more, we cannot simply know if it was used during or after exercise with the title only‘. Is there a reason you are only looking at the title of studies?

Line 856: Again, I disagree with the notion that all aspects of every ERP component should be reported in each study. Studies have hypotheses and objectives for a reason, and we can’t report on everything.
One suggestion that you might consider is for studies to preregister their hypotheses in a database, etc. This way they cannot simply pick and choose results based on how the data looks after the study is done.
Discussion (in general): I would like to see more of these claims (e.g. relation between ERP components and handedness) backed up with evidence and citations.
Also, how do your findings with the bias risk tool affect the conclusions?

---

## Round 0.2 · Major Revisions

Reviewers are still concerned about major shortcoming of the work, please resubmit the work pending major revision.

**Language Note:** The review process has identified that the English language must be improved. PeerJ can provide language editing services - please contact us at [email protected] for pricing (be sure to provide your manuscript number and title). Alternatively, you should make your own arrangements to improve the language quality and provide details in your response letter. – PeerJ Staff

Reviewer 2 ·

Basic reporting

Although improvements have been made to the introduction, it still lacks key papers on the ERP and P300 from well-known authors such as John Polich or Steve Luck. The reference to Woodman 2010 is not an accurate way to reference the first use of ERPs in research - this is simply referencing a narrative tutorial. John Polich even published a paper in 1997 on the effects of exercise and the P300. Therefore the introduction is still missing a sufficient field and context background.

Grammar and spelling mistakes are still present throughout.

Hypotheses and aims were further clarified.

Experimental design

Although you have made an informative table of the difference EEG devices used, there is no comparison on the different technology specs of these systems to compare. Number of electrodes and electrode locations are important. It appears some of these devices may be mobile systems - if so, do they all have electrode Pz? What reference and ground were used? Sampling rate? What type of peak detection algorithm was used? These are all important factors to consider in ERP research that can change the outcome significantly.

Improvements to following PRISMA guidelines are noted and helps to improve the credibility.

Validity of the findings

No comment.

Reviewer 3 ·

Basic reporting

I think the studies reported on in this paper are described quite well. The authors have made some nice improvements since the first version. I would however recommend a thorough spelling and grammar check of the entire document, as there are still some mistakes.

Experimental design

The paper uses well-defined standards to find papers and assess them.

Validity of the findings

I think that the papers reviewed in this study have been well assessed.

Lines 856-858: ‘Our analysis of risk of biases with the Cochrane tool 2 revealed that studies usually donít report an imbalance in signal quality between for the cycling and non-cycling conditions, which make difficult to assess if there are more artefacts the cycling condition.’ Scanlon et al., 2017, 2019 & 2020 all included brief assessments of artefacts between the conditions, including RMS data noise, and trial numbers after artefact rejection. This might be worth looking at.

Additional comments

I think the authors have addressed all of my comments. I just have some small things to point out, that can be addressed.

Abstract (Discussion): I think the first sentence was meant to be deleted here, as the second sentence is the same thing with more detail

Introduction:
line 61: ‘EEG is a technique still in use today, capturing the micro-currents resulting from the electrical activity generated by ionic flows as neurons connect…’
Please cite this information

Line 108: ‘movement’ probably doesn’t need to be capitalized. Also, this sentence would read easier if split into two sentences, with the second sentence starting at ‘with more and more…’

Line 293: could you clarify, why is the MMN a ‘poor signal response’ ?

Line 294: fix the spelling of ‘oddball’

Table 2: looks great. Just fix the spelling of ‘separately’ in the caption.

Also, you can probably write ‘scanlon et al., 2019’, not ‘2019a’, since there are no others with the same author and year listed here.

Table 5: in Scanlon et al., 2019, the participants cycled outside but only sat down (no fixed cycling) inside.

---

## Round 0.3 · Minor Revisions

According with the latest round of revision, a final very minor revision is still requested before the article is recommended for publication.

Reviewer 2 ·

Basic reporting

The citation you need is Sutton et al., 1965. This is the first paper citing the discovery of the P300 using the oddball paradigm. https://pubmed.ncbi.nlm.nih.gov/5852977/

Still minor grammar and spelling mistaies.

Experimental design

No comment

Validity of the findings

No comment

Additional comments

No comment.

---

## Round 0.4 · accepted · Accept

The authors have addressed the reviewer's comment, and there reviewer were minor, therefore a further round of revision was not necessary. The article is ready for publication